# Impact of Hypoxia on Radiation-Based Therapies for Liver Cancer

**DOI:** 10.3390/cancers16050876

**Published:** 2024-02-22

**Authors:** Alexander Villalobos, Jean Lee, Sarah A. Westergaard, Nima Kokabi

**Affiliations:** 1Department of Radiology, University of North Carolina at Chapel Hill, Chapel Hill, NC 27514, USA; nima_kokabi@med.unc.edu; 2Department of Radiology and Imaging Sciences, Emory University, Atlanta, GA 30322, USA; jean.lee2@emory.edu; 3Department of Radiation Oncology, Duke University, Durham, NC 27708, USA; sarah.westergaard@duke.edu

**Keywords:** hypoxia, radiation therapy, liver cancer, hepatocellular carcinoma, yttrium-90 radioembolization, imaging

## Abstract

**Simple Summary:**

The efficacy of radiation-based therapies is negatively impacted by the low oxygen (O_2_) concentration commonly found within solid tumor tissue—a phenomenon known as tissue hypoxia. At baseline, the incidence of cancer that begins in or spreads to the liver is unfortunately very common among oncology patients and can often be accompanied by a poor prognosis. Radiation-based therapies have played an increasing role in the treatment of primary and metastatic liver cancers. Given these reasons, it is of great importance to identify hypoxic tissues within tumors—for its identification can be utilized to optimize treatment algorithms. This review will briefly discuss the relevant impact of hypoxia on tumor tissue response to radiation therapies, with a focus on liver cancers. Emerging imaging modalities and techniques to assess and potentially modulate tissue and tumor hypoxia will be briefly discussed as well.

**Abstract:**

**Background:** Hypoxia, a state of low oxygen level within a tissue, is often present in primary and secondary liver tumors. At the molecular level, the tumor cells’ response to hypoxic stress induces proteomic and genomic changes which are largely regulated by proteins called hypoxia-induced factors (HIF). These proteins have been found to drive tumor progression and cause resistance to drug- and radiation-based therapies, ultimately contributing to a tumor’s poor prognosis. Several imaging modalities have been developed to visualize tissue hypoxia, providing insight into a tumor’s microbiology. **Methods:** A systematic literature search was conducted in PubMed, EMBASE, Cochrane, and Google Scholar for all reports related to hypoxia on liver tumors. All relevant studies were summarized. **Results:** This review will focus on the impact of hypoxia on liver tumors and review PET-, MRI-, and SPECT-based imaging modalities that have been developed to predict and assess a tumor’s response to radiation therapy, with a focus on liver cancers. **Conclusion:** While there are numerous studies that have evaluated the impact of hypoxia on tumor outcomes, there remains a relative paucity of data evaluating and quantifying hypoxia within the liver. Novel and developing non-invasive imaging techniques able to provide functional and physiological information on tumor hypoxia within the liver may be able to assist in the treatment planning of primary and metastatic liver lesions.

## 1. Introduction

Malignancy involvement of the liver is unfortunately a very common occurrence among oncology patients. The liver is one of the most common sites for metastatic disease and is often associated with a poor prognosis and low 5-year survival rates [1,2]. Primary liver cancer, such as hepatocellular carcinoma (HCC), is the fourth most common cause of cancer death worldwide with a 5-year survival rate below 15% among those untreated [3]. 

Over the past few decades, radiation-based therapies have played an increasing role in the treatment of primary and metastatic liver cancers—with timely treatment and achievement of a tumor response being associated with a prolonged overall survival [4,5,6]. Nevertheless, the efficacy of these therapies has been known since the early 20th century to be negatively impacted by the low oxygen (O_2_) concentration commonly found within solid tumor tissues [7]. Tissue hypoxia has been previously shown to not only decrease the radiosensitivity of tissues, but also act as an important physiologic factor that is critical in the modulation of developmental and metabolic pathways within tumor and non-tumor cells [8]. This in turn has also been found to contribute to tumor progression and resistance to drug therapies [7]. Given these reasons, it is of great importance to identify hypoxic tissues within tumors—because their identification can be utilized to optimize treatment algorithms. 

Identification of hypoxia can be very broadly broken down into two categories: Invasive or noninvasive. Until relatively recent times, the most common way to identify tissue hypoxia was via invasive methods such as direct tissue probing to directly measure oxygen content or with direct tissue explant analysis of endogenous hypoxia markers. These methods, however, are not practical nor suitable for use in a real-world clinical setting. To address this, novel non-invasive imaging methodologies, much of them based on PET and MRI modalities, have been developed and will likely take a bigger role in future endeavors to incorporate the impact on hypoxia on the prediction and assessment of treatment response. 

This review will briefly discuss the relevant impact of hypoxia on tumor tissue response to radiation therapies, with a focus on liver cancers. Emerging imaging modalities and techniques to assess and potentially modulate tissue and tumor hypoxia will be briefly discussed as well. 

## 2. Hypoxia in Tumor Microenvironment

### 2.1. What Is Hypoxia?

Because the identification and measurement of tissue hypoxia has been historically performed with oxygen probes, tissue hypoxia is usually defined in terms of partial pressure of O_2_ within a tissue (i.e., pO_2_). While different groups have defined hypoxia differently, the general threshold definition of tissue hypoxia is pO_2_ <20 mmHg—with moderate and extreme tissue hypoxia defined as pO_2_ < 10 mmHg and <1 mmHg, respectively [9]. While these rudimentary pO_2_ cutoffs attempt to simplify what it means to classify a tissue as hypoxic, the metabolic process and temporospatial state of hypoxia is much more complex. 

In the liver, the hypoxic state can arguably be more complex than in other parts of the body because of the parenchymal fibrotic changes that occur as a result of cirrhosis. Specifically, the liver fibrosis that occurs as cirrhosis worsens increases the resistance to blood flow (i.e., portal hypertension), which together with sinusoidal capillarization, sinusoidal obstruction syndrome, and variceal associated decreased portal venous flow can worsen liver tissue hypoxia [10]. This state of worsened liver tissue oxygenation in the setting of cirrhosis creates a hypoxic tissue microenvironment that has been associated with the promotion of liver tumor growth [11]. This hypoxic tissue microenvironment is often made even worse (i.e., more hypoxic) within the HCC tumor itself because of its relative rapid growth [12], thereby making HCC one of the most hypoxic malignancies [13]. 

### 2.2. Hypoxia Pathogenesis and Associated Metabolic Reprogramming

Localized tissue hypoxia can result from two types of insufficient oxygen availability. Acute hypoxia, also known as perfusion-limited hypoxia, is caused by a temporary reduction in blood supply to a tissue, arising from either structural or functional abnormalities of the tumor’s micro vessels. Alternatively, chronic hypoxia, also known as diffusion limited hypoxia, is caused by an increase in diffusion distance (>70 μm) within the tissue. Regardless, any form of hypoxia promotes changes in a cell’s microenvironment. 

In tumors, hypoxic cells adapt to oxygen starvation to preserve both the tumor’s growth and propagation. A classic example of this is the hypoxic cell’s ready switch from aerobic respiration to anaerobic glycolysis, which increases the consumption of glucose and the production of pyruvate (which is then subsequently converted to lactate). Even in the presence of oxygen, these cancer cells can have a preference for metabolic pathways often associated with a hypoxic environment (i.e., Warburg Effect [14]). This sort of metabolic reprogramming is also accompanied by proteomic and genomic transformations that are predominantly regulated by hypoxia-inducible factors (HIFs). HIFs are a family of transcription factors that prove critical to a cell’s response to hypoxia by activating the transcription of a myriad of genes. These transcriptions can in turn mediate a tumor cell’s motility, epithelial–mesenchymal transition, extracellular matrix remodeling, glucose and lipid metabolism, immune evasion, stem cell specification, surrounding angiogenesis, and even invasion and metastasis to local and distant tissues [15]. While similar results have been shown with other non-liver malignancies [15], increased HIF expression has been associated with an increased incidence of macrovascular invasion and a decreased incidence of disease-free and overall survival among patients with HCC [16]. 

## 3. Impact of Hypoxia on Radiation-Based Therapies

### 3.1. Basics of Radiation Damage and Oxygen’s Direct Role

The biologic effects of radiation primarily result from damage to a cell’s deoxyribonucleic acid (DNA) [17]. This radiation-associated damage can occur in either a direct or indirect manner. In the direct radiation damage method, DNA damage occurs as a direct result of the radiation. Oxygen’s presence does not impact the degree of DNA damage that immediately occurs as a result of direct radiation damage alone. Of note, direct radiation damage is the dominant DNA damage process for high linear energy transfer (LET) (e.g., neutrons or alpha particles) radiation. In the indirect radiation damage method, radiation primarily interacts with molecules (e.g., water) in the cell to produce free radicals that then go on to inflict DNA damage. The lifetime of these free radicals is extremely short [17]. However, the damage that they cause is estimated to account for two-thirds of the DNA damage that is caused by low-LET radiation methods. Furthermore, the presence of oxygen takes on an important role in the indirect radiation damage method by ‘fixing’ (i.e., make permanent) the DNA damage inflicted by the free radicals (i.e., oxygen fixation hypothesis [17]). 

### 3.2. Hypoxia’s Impact on Radiation-Based Therapies

Classically, the radio resistance afforded by hypoxia is based on the oxygen fixation hypothesis, where the presence of molecular oxygen during radiation exposure fixes, or makes permanent, the DNA damage produced by free radicals—thereby improving the efficacy of radiation therapies [17]. This, in essence, means that the dose of radiation needed to obtain the same cytotoxic biological effect is higher in hypoxic tissues than in normoxic tissues, with some studies reporting an oxygen enhancement ratio of up to 2.5 to 3.5 [18]. 

Hypoxic stress also induces proteomic and genomic changes that are pro-tumor and largely regulated by HIF proteins [19]. For example, hypoxia-induced glucose metabolism reprogramming has been associated with the increased production of antioxidants capable of neutralizing DNA-damaging free radicals created by radiation therapy [20]. Since radiation therapies are known to have a markedly increased kill efficacy on proliferative cells, the HIF-mediated cell cycle retardation and arrest have been associated with a decreased efficacy of radiation treatments [20]. By promoting the production of proangiogenic cytokines (e.g., vascular endothelial growth factor (VEGF)), HIF proteins are also able to provide a protective effect to radiated tumor blood vessels [20]. These in turn permit radioresistant hypoxic tumor cells to then become re-oxygenated after the death of the radiated tumor cells, and then translocate towards blood vessels capable of carrying them on to create tumor recurrence and/or metastatic disease [21]. Lastly, hypoxic tissue regions are known to accumulate and propagate cancer stem cells (CSC). In addition to having multiple pro-tumor phenotypic characteristics, these CSC promote tumor radiotherapy-resistance through multiple pathways, including through the increased activation of DNA repair pathways [22] and the relatively lower level of endogenous free radicals within the CSC [23]. 

## 4. Methods to Measure Hypoxia

Given the detrimental effects of hypoxia within tumors and its role in resistance to radiation and systemic therapies, extensive efforts have been carried out to quantify and assess tissue hypoxia. To date, methods to assess tumor hypoxia can be separated into three major groups: methods that directly assess oxygen concentration, methods that measures the physiologic process of oxygen molecules, and methods that evaluate endogenous markers as a response to hypoxia. A brief description of these is found below (Table 1). 

### 4.1. Direct Methods

#### 4.1.1. Oxygen Electrode

The widely accepted gold standard for quantifying tissue is by the direct pO_2_ measurement with an invasive polarographic electrode. These types of probes measure oxygen from several points per needle track by being inserted into a tissue. The average pO_2_ measurement is then considered the tissue’s overall hypoxia status. Because this technique involves physically inserting a probe into a tissue, this technique is prone to sampling error, inter-operator variability, and limited to relatively easily accessible tumors [19]. Furthermore, the probe does not discriminate between viable and necrotic tissue—possibly overestimating hypoxia when necrotic areas are sampled. Because the spatial resolution is of roughly 50–100 cells adjacent to the probe, the construction of three-dimensional oxygen maps (and thus the utilization of this technique for meaningful therapy planning) is very limited. Few studies have utilized oxygen electrodes for the direct assessment of liver tumor hypoxia in rat models [24]. While HCC is widely accepted as a very hypoxic tumor, direct measurement data (particularly with oxygen electrodes (i.e., the gold standard) of HCC’s hypoxia status remain lacking [24]. 

#### 4.1.2. Phosphorescence Quenching

Phosphorescence quenching enables visualization of perfused vessels in tumors. Specifically, oxygen-sensitive phosphorescent molecules are invasively introduced to the tissue (either via a solution or a physical probe) and then illuminated with a short flash of light which then causes the molecules to emit their own light at an intensity that decays exponentially at a rate proportional to the local oxygen concentration. Using pre-calibrated parameters that include the Stern–Volmer constant and the rate of decay in the absence of oxygen, the rate of light intensity decay translates into tissue oxygen concentration. Because phosphorescence quenching is an invasive method, it also shares the same limitations as the oxygen electrode method. To date, its use for the evaluation of hypoxia in human hepatic malignancies remains limited [25]. 

#### 4.1.3. Electron Paramagnetic Resonance

Electron paramagnetic resonance (EPR) requires the injection or implantation of an exogenous probe bearing an unpaired electron that is selective in its interaction with oxygen. The width of the spectral band corresponding to the injected exogenous probe signal, as measured by an EPR imager (a type of MRI device [58]), correlates with oxygen concentration and provides quantitative pO_2_ values. Because the injected exogenous probe is metabolically inert, this EPR method can assess the oxygenation of a tissue from a few minutes to even months—permitting a nuanced analysis of the tissue’s oxygenation status over time. Additionally, this technique can be co-registered with anatomic imaging data (such as from MRI) to then provide a three-dimensional anatomical picture of a tissue’s oxygenation. To date, this type of hypoxia assessment method remains predominantly limited to pre-clinical studies [26]. 

#### 4.1.4. Oxygen-Sensitive Contrast Magnetic Resonance

By utilizing perfluorocarbon-based compounds (PFCs) and OX63 (a paramagnetic oxygen-sensitive triaryl methyl radical contrast) as the injected contrast agents, ^19^F-magnetic resonance spectroscopy, and Overhauser-enhanced magnetic resonance imaging can report oxygen concentrations within the tissue regions of interest containing the contrast agents. When injected systemically, these contrast agents have demonstrated a deposition profile that involves the liver [27,28], which could in theory create some difficulty evaluating liver tumors using these agents. The blood clearance half-life of PFCs ranges from 3 to 42 h [58], permitting a more limited time frame of hypoxia imaging than EPR. This type of hypoxia assessment method remains predominantly limited to pre-clinical studies [19]. 

### 4.2. Endogenous Markers of Hypoxia 

#### 4.2.1. Immunohistochemical Staining 

Immunohistochemical staining (IHC) is a technique where antigens (proteins) in cells of a tissue section are ‘stained’ by the selective binding of an antibody attached to a reporter/label molecule (e.g., fluorescent dyes, enzymes, radioactive elements). Utilizing these techniques, hypoxia-associated factors have been explored, including HIF-1α and HIF-2α. Because HIF-1/2α proteins have a short half-life upon re-exposure to oxygen and are usually localized in the nucleus under hypoxic conditions, IHC for these proteins requires nuclear permeabilization and can thus preclude a doable but challenging IHC target. As a result of this, some researchers have focused on targeting more abundant and easier to immunolabel HIF-transcriptional targets such as glucose transporter 1 (GLUT-1), monocarboxylate transporter 1 (MCT-1), and other downstream target genes of HIF. In the liver, the activation of HIF proteins and its downstream target genes has been shown to have a time-sensitive and disease-dependent dichotomous role that fluctuates from protective to detrimental depending on the acuteness of the liver injury. Specifically, HIF proteins provide a hepato-protective function in the setting of acute liver damage by promoting the protection of cells, promotion of angiogenesis, and reprogramming of cellular energy metabolisms [29]. However, it is these same functions that promote a pathological role of HIF in fibrogenesis, hepatic lipid accumulation, and tumor progression during chronic liver diseases [30]. Regarding the limitations of the IHC technique, it cannot be performed in real-time because it requires tissue preparation and staining. Additionally, results are agnostic to other non-hypoxia mechanisms that are known to influence HIF and downstream protein expression [31]. 

#### 4.2.2. Comet Assay 

Single-cell gel electrophoresis assay, also known as comet assay, is a technique that can indirectly estimate hypoxia in radiated solid tumors by measuring DNA strand breaks. This technique works by first embedding treated cells in agarose. These are then lysed, electrophoresed, and stained with fluorescent DNA dye. During electrophoresis, more of the broken DNA migrates into the “tail” of the electrophoresis gel compared to intact DNA. This comet “tail” is then calculated to estimate the amount of DNA damage. Because it has been previously shown that oxygenated cells are 3x more sensitive to low-LET radiation than hypoxic cells, the amount of DNA damage can then be utilized to estimate the degree of hypoxic cells [19]. Since this method strictly measures the degree of DNA damage, it can also be utilized to evaluate the effect of non-radiation therapies on the tissue of interest. In the liver, this method has been utilized to show that hypoxia reduces the efficacy of radiation and chemotherapies [32,33]. Like the IHC technique, the comet assay technique cannot be performed in real-time and can also be prone to detecting DNA damage caused by unanticipated factors. 

### 4.3. Physiologic Methods

#### 4.3.1. Near-Infrared Spectroscopy/Tomography

Near-infrared spectroscopy (700 to 900 nm wavelengths) relies on the different absorption spectra of hemoglobin (Hb) and oxy-hemoglobin (HbO_2_) to quantify a ratio of Hb/HbO_2_. This ratio can then be extrapolated into pO_2_ with the utilization of the hemoglobin saturation curves. This method does not directly measure oxygen concentrations. Because this method requires light penetration and capturing, it is significantly limited to tissues with low light attenuation and of limited size. In the liver, this technique has been shown in animals to be able to measure hepatic oxygenation whenever the devices are surgically positioned directly on the liver surface [34]. However, these devices have failed to accurately measure hepatic oxygenation whenever they are placed on patient’s skin [35]. 

#### 4.3.2. Photoacoustic Imaging

Optoacoustic imaging, also known as photoacoustic imaging, is a technique where a light-emitting source (e.g., laser) is utilized to radiate a tissue. This tissue absorbs the light and converts the energy into heat, which through transient thermoelastic expansion creates wideband ultrasound waves that can be captured and analyzed. Because the characteristics of the ultrasound wave correlate to the degree of optical absorption (which is known to be closely associated with physiological properties (such as that of endogenous HbO_2_ and Hb)), this method can indirectly extrapolate the estimated degree of oxygen concentration. Because this technique is fundamentally an ultrasound technique, it has a high spatial resolution and relatively deep tissue penetration that can also be reconstructed in three-dimensional space [36]—thereby providing promise in its utilization for hypoxia assessment in vivo. However, this technique remains predominantly limited to animal studies (which themselves have shown great difficulty attaining accurate oxygen concentration evaluations with the greater depths that would be required to assess liver tumor hypoxia in humans) [37]. 

#### 4.3.3. Contrast-Enhanced Ultrasonography

Ultrasound contrast agents are predominantly composed of microbubbles, which themselves are made of inert high-molecular-weight gases stabilized by a shell of composed of surfactants, phospholipids, polymers, or proteins [38]. Commercially available ultrasound contrast agents are usually small (1–8 um) and injected intravenously. As such, they readily pass through the pulmonary bed and reach the systemic circulation. Compared with the surrounding tissue, the gas contained within the microbubbles affords them a different acoustic impedance and compressibility, thereby permitting ultrasound receivers to discern the microbubbles within the vasculature (i.e., assess blood flow). While it has not been directly validated in the liver, this technique’s ability to assess tissue perfusion has been correlated with tissue hypoxia in the head and neck space [19]. As such, relative areas of tissue hypoxia can in theory be deduced in real time by identifying differential tissue perfusion within liver tumors. Contrast-enhanced ultrasonography has demonstrated promise in its utilization for the prediction and assessment of liver tumor response after therapy [39]. Additionally, there are rapidly emerging techniques that utilize the injected microbubbles as either a carrier or therapeutic agent capable of facilitating the treatment and/or radio sensitization of tumors [40].

#### 4.3.4. Position Emission Tomography

Non-invasive hypoxia imaging with positron emission tomography (PET) relies on the detection of radiolabeled tracers that are often systemically introduced to facilitate the detection of tumor hypoxia via the radiolabeling of hypoxic cells. Several PET radiotracers have been developed for the study of tumor hypoxia, including copper-, gallium-, iodine-, and technetium-based PET agents [41]. Nevertheless, most of the classically utilized radiotracers for clinical and research applications are based on the Nitroimidazole family of compounds (e.g., ^18^Fluoromisonidazole (^18^F-MISO), 1-(5-fluoro-5-deoxy-α-D-arabinofuranosyl)-2-nitroimidazole) (^18^F-FAZA), ^18^Ffluoroetanidazole (^18^F-FETA)) [41], which radiolabel hypoxic cells by having their nitroimidazole isotopes irreversibly bind to thiol groups on metabolic proteins at rates inversely proportional to the cell’s oxygen concentration [42]. Utilization of these radiotracers for the study of hypoxia in liver tumors remains very limited [41], in part because of the difficulties associated with imaging liver tumors on the background of the relatively hypoxic (and often fibrotic/cirrhotic) liver parenchyma that is often radiolabeled at baseline by the many radiotracers whose metabolism and/or deposition often involves the liver. In the few studies that have looked at PET radiotracers for human liver tumors, ^18^F-MISO PET has been found to be capable of assessing tumor response after catheter-directed and systemic therapies [43,44]. Clinical trials evaluating the capability of PET radiotracers to predict and assess liver tumor response to therapies are ongoing [45,46].

#### 4.3.5. Dynamic Contrast-Enhanced Magnetic Resonance Imaging

Perfusion data obtained with dynamic contrast-enhanced magnetic resonance imaging (MRI) (DCE-MRI) and a gadolinium-based contrast agent, such as Gd-diethylenetriaminepentaccetic acid (Gd-DTPA), has been associated with tissue oxygenation [47]. Specifically, hydrophilic small molecule contrast agents like Gd-DTPA have been shown to diffuse past blood vessel walls and distribute within a tumor’s extracellular space as a function of blood perfusion, vascular density, tissue permeability, and extracellular volume fraction [48]. In other words, tumor Gd-DTPA T1 enhancement, including for liver tumors, is indirectly associated with increased tumor tissue oxygenation and can be used as a prognostic/assessment tool for treatment [40,49].

#### 4.3.6. Special Magnetic Resonance Sequences

While functional imaging of hypoxia with MRI techniques has had many exciting developments [50], classical non-invasive MRI hypoxia imaging methods have primarily focused on the dioxygen molecule (O_2_) in solution and the deoxyhemoglobin monomer (Hb). These two molecules have magnetic properties that can be exploited so that they can be visualized with specialized MRI sequences. 

Blood-oxygen-level-dependent (BOLD) MRI sequences exploit deoxyhemoglobin’s paramagnetic properties (oxyhemoglobin does not have a magnetic moment) to visualize deoxyhemoglobin’s relative concentration within tissues. This permits BOLD MRI to indirectly assess the oxygenation of a tissue—a finding that has been validated with other methods of tissue hypoxia measurement [51]. Pre-therapy BOLD MRI has been shown to be a capable baseline imaging biomarker able to predict treatment outcomes in non-liver human malignancies [52,53]. Results demonstrating BOLD MRI’s capability to predict and assess treatment response in human liver tumors remain limited [54,55]. Clinical trials evaluating the utility of BOLD MRI to predict and assess human liver tumor response radiation treatments are ongoing [46].

Tumor-oxygenation-level-dependent (TOLD) MRI sequences exploit the paramagnetic properties of free O_2_ molecules in the tissue. Specifically, TOLD MRI employs the use of 100% inhaled O_2_ to induce arterial hyperoxia to disrupt tumor concentrations of oxygen molecules in solution, thereby inducing magnetic changes that can be used to extrapolate the degree of tissue hypoxia. Studies evaluating the ability of TOLD MRI to assess human liver tumor hypoxia are limited [56].

## 5. Potential Strategies to Overcome Hypoxia

With the goal of improving tumor response to radiation and chemotherapies, extensive efforts have been made to target tumor hypoxia in both clinical and experimental settings for over a century. Generally speaking, these efforts have been focused in either directly increasing the overall oxygen concentration in tumors (e.g., carbogen, hyperthermia, hyperbaric oxygen), indirectly increasing tumor oxygen concentration by decreasing oxygen consumption in tumor adjacent healthy tissue (e.g., metformin, hyperthermia, atovaquone), directly targeting hypoxic tumor cells with hypoxia-activated prodrugs that can be cytotoxic and/or radio-sensitizing (e.g., nimorazole, tirapazamine, SN30000), and directly targeting HIF proteins to inhibit hypoxia-driven radiation insensitivity and cancer progression. A more detailed review of these hypoxia-targeting efforts is reviewed elsewhere [57].

## 6. Conclusions

Tumor hypoxia is unfortunately often present in malignant solid tumors, creating a well-established biological phenomenon that negatively impacts the prognosis of malignancies regardless of chosen treatment modality. Hypoxia-based patient stratification has yet to be readily adopted into clinical practice. However, it is well recognized that hypoxia assessment can be predictive of outcomes and help identify high-risk patients in need of treatment modification. While extensive efforts have been previously made to target tumor hypoxia, it is only in relatively recent times that non-invasive hypoxia imaging methods have evolved, with liver-specific hypoxia imaging methods exhibiting a relative paucity of research effort as compared to malignancies in other non-liver body parts. Whether in combination or individually, these evolving hypoxia imaging modalities can and will continue to provide functional and physiological information on tumor hypoxia. This, in return, will provide invaluable insight and predictive value capable of improving the prognosis of oncologic patients.

## Figures and Tables

**Table 1 cancers-16-00876-t001:** Brief description of the methods available to measure hypoxia.

Modality		Brief Description	References
**Direct** **Measurement**	Oxygen Electrode	Utilizes an invasively introduced electrode to directly measure partial pressure of oxygen within a tissue.	[19,24]
	Phosphorescence Quenching	Utilizes invasively introduced oxygen-sensitive phosphorescent molecules that, when illuminated, emit their own light at a rate proportional to the local tissue oxygen concentration.	[19,25]
	Electron Paramagnetic Resonance	Utilizes an invasively introduced (injected vs. implanted) exogenous probe that produces a signal, as measured by magnetic resonance imaging, that correlates to the local tissue oxygen concentration. Hypoxia imaging can be performed with a wide temporal range (up to multiple years).	[19,26]
	Oxygen-Sensitive Contrast Magnetic Resonance	Utilizes an invasively introduced (injected) perfluorocarbon-based compound that acts as an injected contrast, able to be visualized by magnetic resonance imaging, that is sensitive to the local tissue oxygen concentration. Hypoxia imaging can be performed with a more limited temporal range (up to a month or two).	[19,27,28]
**Endogenous Hypoxia Markers Measurement**	Immunohistochemical Staining	Utilizes an antibody attached to a reporter/labeled molecule to ‘stain’ and image hypoxia-associated antigens (markers) in cells of a tissue section.	[19,29,30,31]
	Comet Assay	Utilizes a single-cell gel electrophoresis assay to indirectly estimate hypoxia in radiated solid tumors by measuring DNA strand breaks.	[19,32,33]
**Physiologic Hypoxia Measurement**	Near-Infrared Spectroscopy/Tomography	Utilizes near-infrared spectroscopy to quantify a ratio of hemoglobin/oxy-hemoglobin within a tissue. This ratio is then utilized to extrapolate the partial pressure of oxygen within the illuminated tissue.	[19,34,35]
	Photoacoustic Imaging	Utilizes a light-emitting sources (e.g., Laser) to radiate a tissue and cause thermoelastic expansion capable of creating ultrasonic waves that can then be analyzed to indirectly extrapolate the estimated oxygen concentration within a tissue.	[19,36,37]
	Contrast-Enhanced Ultrasonography	Utilizes systemically injected microbubbles to assess perfusion within a tissue, which can then be correlated with tissue hypoxia.	[19,38,39,40]
	Positron Emission Tomography	Utilizes systemically injected radiotracers capable of radiolabeling hypoxic cells within a tissue. Once radiolabeled, hypoxic cells can then be imaged and analyzed with positron emission tomography (PET).	[19,41,42,43,44,45,46]
	Dynamic Contrast-Enhanced Magnetic Resonance Imaging	Utilizes systemically injected magnetic resonance imaging contrast agents to attain perfusion data within a tissue, which can then be correlated with tissue hypoxia.	[19,47,48,49,50]
	Special Magnetic Resonance Sequences	Multiple special magnetic resonance sequences exist. In general, these special sequences exploit the paramagnetic properties of oxygen molecules and/or tissue proteins impacted by the presence of oxygen. By directly observing these, the degree of local tissue oxygen concentration can be estimated.	[19,51,52,53,54,55,56,57]

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
