# Peer review of "Impact of Hypoxia on Radiation-Based Therapies for Liver Cancer"

_cancers, 2024, doi:10.3390/cancers16050876_

Round 1

Reviewer 1 Report

Comments and Suggestions for Authors

The relevance of tumour hypoxia in general and its regional distribution in and around the tumour cannot be overestimated when it comes to understanding the tumour microenvironment. Since it has an effect on all forms of therapy - symmetric therapy, radiotherapy, ablation and endovascular therapies - it is all the more surprising that this aspect is still insufficiently researched.
The present summary of the various procedures is comprehensive, and appropriate in length.
It is time to advance hypoxia-based patient stratification in multimodal, interdisciplinary oncology. This review makes a relevant contribution to this.

Reviewer 2 Report

Comments and Suggestions for Authors

The article reviewed the impact of hypoxia on radiation-based therapies for liver cancer. Overall, the article sounds fascinating, but my comments need to be addressed in the text. For the following reasons, I conclude that the paper should undergo major revisions.

1-      The present review article does not have sufficient literature that can be considered for review.

2-      There are too many errors in reference numbering, for example, the authors used only 58 references in the text, but the numbers in the references list are duplicated.

3-      The present review article does not have sufficient literature that can be considered for review.

4-      In the abstract and the introduction, clarifying the aim of the study is required.

5-      I recommended that a brief explanation of different imaging modalities, that are important for liver cancer is needed.

6-      There are no tables or images to simply show the authors' claims about different imaging modalities. I strongly recommend this important issue be added to the text.

7-      There is no conclusion section in the text and this is very important in the review article, discussion should be provided as concrete on the text.

8-      The conclusion is too long and a paragraph should be provided as concluding remarks at the end of the text.

9-      Grammatical and sentence errors in the article, and the language organization needs to be improved.

Comments on the Quality of English Language

Grammatical and sentence errors in the article, and the language organization needs to be improved.

Reviewer 3 Report

Comments and Suggestions for Authors

In this paper, the authors conducted a systematic literature search in PubMed, EMBASE, Cochrane, and Google Scholar for all reports related to hypoxia on liver tumors.  The authors summarized all relevant studies.

The authors emphasized the impact of hypoxia on liver tumors and reviewed PET-, MRI-, and SPECT-based imaging modalities that have been developed to predict and assess a tumor’s response to radiation therapy – with a focus on liver cancers.

The authors conclude that while numerous studies have evaluated the impact of hypoxia on tumor outcomes, there remains a relative paucity of data evaluating and quantifying hypoxia within the liver.

The authors state that novel and non-invasive imaging techniques able to provide functional and physiological information on tumor hypoxia within the liver may be able to assist in the treatment planning of primary and metastatic liver lesions.

Overall, this is an interesting study on a topic of great interest. I commend the authors on this interesting paper.

Round 2

Reviewer 2 Report

Comments and Suggestions for Authors

Dear Authors

According to my comments, the authors responses to them and I think, the table need to revision in point of references view. Please put the references as a individual column in the table.

Author Response

Reviewer 2:

According to my comments, the authors responses to them and I think, the table need to revision in point of references view. Please put the references as a individual column in the table.

We thank the reviewer for their feedback. I have attached the revisions requested, including adding a reference to the table as well as adding the citations per comment.  
